# RAFT: A Real-World Few-Shot Text Classification Benchmark

**Neel Alex**[*]
Ought

**Eli Lifland**[*]
Ought

**Lewis Tunstall**
Hugging Face

**Abhishek Thakur**
Hugging Face

**Pegah Maham**
Stiftung Neue Verantwortung

**C. Jess Riedel**
NTT Research

**Emmie Hine**
Oxford Internet Institute

**Carolyn Ashurst**
Alan Turing Institute

**Paul Sedille**
Harvard Kennedy School

**Alexis Carlier**
University of Oxford

**Michael Noetel**
Australian Catholic University

**Andreas Stuhlmüller**
Ought

## Abstract

Large pre-trained language models have shown promise for few-shot learning, completing text-based tasks given only a few task-specific examples. Will models soon solve classification tasks that have so far been reserved for human research assistants? Existing benchmarks are not designed to measure progress in applied settings, and so don't directly answer this question. The RAFT benchmark (**R**eal-world **A**nnotated **F**ew-shot **T**asks) focuses on naturally occurring tasks and uses an evaluation setup that mirrors deployment. Baseline evaluations on RAFT reveal areas current techniques struggle with: reasoning over long texts and tasks with many classes. Human baselines show that some classification tasks are difficult for non-expert humans, reflecting that real-world value sometimes depends on domain expertise. Yet even non-expert human baseline F1 scores exceed GPT-3 by an average of $0.11$. The RAFT datasets and leaderboard will track which model improvements translate into real-world benefits at https://raft.elicit.org.

## 1   Introduction

Few-shot learning, the capacity to complete a task given a small number of demonstrations [11], is one of the hallmarks of human intelligence [30, 17]. As researchers, we leverage this capacity when we delegate work on crowdsourcing platforms or give a task with examples to a human research assistant.

Brown et al. [6] show that large pre-trained language models exhibit few-shot learning capabilities for a wide range of natural language tasks. If those capabilities were comparable to people on economically relevant tasks, this would be important to know: a single model could be used across multiple real-world tasks, with low per-task data labeling cost. However, these models have also been shown to have inconsistent few-shot performance depending on the exact setup and task being solved [e.g. 21, 24]. The mixed evidence suggests that it would be valuable to measure and track few-shot performance on a set of tasks that is representative of what appears in practice.

---

[*]Equal contribution. Correspondence to elifland@ought.org and salexucb@berkeley.edu. NA contributed during an internship at Ought.

35th Conference on Neural Information Processing Systems (NeurIPS 2021) Track on Datasets and Benchmarks.

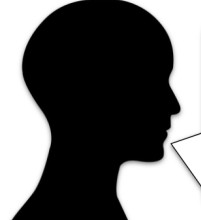

The dataset is a list of institutions that have contributed papers to semiconductor conferences in the last 25 years, as catalogued by IEEE and sampled randomly. The goal is to classify the institutions into one of three categories: "university", "company" or "research institute". 50 labeled examples are provided.

Task Specification in Natural Language

| Organization | Paper Title | Label |
|---|---|---|
| North Carolina State Univ.,Raleigh,NC,USA | 3Gb/s AC-coupled chip-to-chip communication using a low-swing pulse receiver | university |
| Advanced LCD Technology Development Center... | Sub-Micron CMOS / MOS-Bipolar Hybrid TFTs for System Displays | company |
| imec,Heverlee,Belgium | 24.4 A 680nA fully integrated implantable ECG-acquisition IC with analog... | research institute |
| ... | ... | ... |

Partially-labeled Dataset

Label the sentence based on whether it is related to an adverse drug effect (ADE)

The following is a banking customer service query. Classify it as one of the following 77 classes.

Label the impact statement as "mentions a harmful application" or "doesn't mention a harmful application" based on whether it mentions a harmful application of the research done in the paper.

Identify whether this paper should be included in a meta-review which includes the findings of systematic reviews on interventions designed to promote charitable donations.

...

Other Tasks

Figure 1: RAFT includes naturally occurring classification datasets, mimicking work that is usually given to human research assistants. Each task comes with natural language instructions and labels in addition to 50 training examples.

Natural language tasks coarsely split into generation, classification, and retrieval. We focus on classification tasks because they support high-quality automated evaluation, cover a wide range of economically valuable tasks, and yet don't have existing real-world benchmarks.

Existing few-shot classification benchmarks are typically designed to highlight areas where models fall short [29] or to study particular model abilities [5, 37, 21]. The tasks and evaluation setup aren't optimized to measure progress in applied settings:

- *Tasks* that are generated or chosen specifically to test language models may not represent some of the challenges found when applying these models in real-world settings. For example, SuperGLUE [32] and the few-shot equivalent FewGLUE [29] mainly include short texts. Doing well on applied tasks sometimes requires reasoning over long texts. Existing systems struggle with long texts due to a limited context window, especially in the few-shot setting where some systems learn from examples presented in context.

- The *evaluation* does not closely mirror deployment, and may both under- and overestimate models' capabilities. It may underestimate model capability by restricting models to the closed-book setting (e.g., no retrieval from online sources) and using uninformative labels (e.g., 0/1 instead of "about literature" vs. "about movies"). It may overestimate model capability by using many more than a few examples for setting hyperparameters during validation [24].

RAFT is a real-world few-shot text classification benchmark designed to measure how much recent and upcoming NLP advances benefit applications:

- The *tasks* are naturally occurring tasks. Their labeling is inherently valuable to someone, and they may have challenges that are not reflected in synthetic tasks. Inherent value means that, if it were sufficiently fast and cheap, it would be desirable to outsource the task to human research assistants or crowd workers. Challenges refers to the need for information retrieval, domain expertise, parsing long documents, and making use of instructions. Table 1 shows the real-world challenges presented by RAFT, including 4 datasets with long input texts.

- The *evaluation* closely mirrors deployment. For each task, we release a public training set with 50 examples and a larger unlabeled test set[2]. We encourage unsupervised pre-training on the unlabelled examples and open-domain information retrieval. We keep the test-set labels private and provide automated evaluation through a Hugging Face leaderboard[3].

In addition to the gold-standard human labels, we collect automatic and crowdsourced baselines. The automatic baselines reveal areas where current techniques struggle, such as reasoning over long texts and tasks with many classes. The crowdsourced baseline reveals that RAFT includes a mix of moderate to difficult tasks. We also observe difficulties in collecting human crowdsourced baselines on some datasets, particularly when domain expertise is important, which suggests that real-world value often depends on domain knowledge.

The RAFT datasets and leaderboard can be viewed and submitted to at https://raft.elicit.org.

## 2 Related Work

We briefly review few-shot learning in NLP, then the benchmarks that are most similar to RAFT.

### 2.1 Few-shot learning in NLP

Pre-trained language models (PLMs) such as BERT [10] and GPT-3 [6] can learn to do some NLP tasks when prompted with a few demonstrations, including some classification tasks. The two primary approaches to few-shot classification using PLMs are in-context learning and prompt-based fine-tuning.

**In-context learning.** A PLM is primed with labeled examples in its prompt. It classifies the example included at the end of the prompt by predicting the classification conditioned on the priming. GPT-3 [6] used in-context learning to achieve promising results on a variety of classification tasks. UniFew [5] similarly achieved strong results on classification tasks via in-context learning, converting classification tasks into a multiple-choice question answer format for prompting.

**Prompt-based fine-tuning**. A PLM is fine-tuned with masked-language modeling objectives to learn from few examples. This is also known as Pattern-exploiting training (PET) [28]. While PET requires task-specific prompts, it achieves better performance than GPT-3 in-context with smaller models [29]. LM-BFF [13] improves prompt-based fine-tuning by dynamically constructing prompts.

### 2.2 Few-shot NLP benchmarks

The most closely related few-shot NLP benchmarks are FLEX [5], FewGLUE [29], CrossFit [37], and NaturalInstructions [21]. Each of these benchmarks includes at least some classification tasks with meaningful textual labels.

These benchmarks are designed to study transfer between tasks [5, 37], pinpoint where NLP models fall short [29], and evaluate ability of models to follow instructions [21], whereas RAFT is designed to be representative of real-world classification tasks. This difference in goals is reflected in selection of tasks and evaluation:

**Tasks.** FLEX, FewGLUE and NaturalInstructions test on traditional NLP tasks. CrossFit tests on 160 tasks from the Hugging Face Datasets[4] and includes some naturally occurring datasets, including the TweetEval dataset [2] that RAFT uses as well. CrossFit excludes tasks that leverage external sources or information retrieval techniques, need domain knowledge (e.g. COVID-19 datasets), and long documents (e.g. scientific papers). Like RAFT, FLEX includes tasks with strong class imbalance.

**Evaluation.** None of the existing benchmarks allow open-domain information retrieval. Like FLEX, RAFT provides no extra validation data beyond the training examples. Perez et al. [24] argue that the performance of state-of-the-art few-shot methods has been overestimated by most other existing benchmarks because they use labeled examples beyond the few training instances provided for model and parameter selection.

---

[2]Datasets are at https://raft.elicit.org/datasets
[3]Instructions for submission are at https://raft.elicit.org/submit
[4]https://huggingface.co/datasets

# 3 Benchmark Description

RAFT is a few-shot *classification* benchmark. We focus on classification primarily because automatic evaluation is more reliable than for generation tasks. We believe (as our results will later confirm) that there still is a substantial gap between even non-expert humans and automated systems in the few-shot classification setting.

Both tasks (datasets and metadata) and evaluation (rules for submission, metrics) are chosen to mirror real-world classification problems.

## 3.1 Tasks

A classification task is a dataset with labeled natural language entries. Each label corresponds one-to-one with a natural language class name. Each task has instructions for labeling.

### 3.1.1 Dataset selection criteria

We selected datasets based on the following criteria ("non-trivial real world tasks"):

**Naturally occurring.** We focus on data that are naturally occurring, rather than being synthetically generated to test and improve language models.

**Intrinsic value.** We select datasets for which the correct labeling inherently provides real-world value. RAFT includes tasks like hate-speech detection, medical case report parsing, and literature review automation, where better performance translates into practical benefits. This criterion involves subjectivity, but we aimed to select tasks that approximate the distribution of valuable classification tasks well.

**Realistic class distribution.** We did not exclude datasets with heavily imbalanced classes.

**Open-domain feasibility.** As we provide an open-domain setting where information retrieved from the web may be used to augment predictions, we excluded tasks for which the correct label is extremely easily discoverable through a Google search. For example, we considered including the LIAR [35] dataset which includes Politifact statements and their veracity. We decided against including it since it would be trivial to get 100% accuracy by running a site search on https://www.politifact.com/.

In order to gather datasets meeting the above requirements, we put out a collaboration request. We also reached out to users of classification on Elicit [23]. Lastly, we conducted a search of existing datasets on the Hugging Face Hub[5] and PapersWithCode [6].

### 3.1.2 Dataset preparation

In cases where the test set was over 5,000 data points, we randomly selected 5,000 to serve as a test set in order to keep the test set sizes manageable. When the dataset didn't already have textual labels, we added textual labels according to our best understanding of the task.

### 3.1.3 Selected RAFT datasets

We selected 11 datasets, accessible at https://raft.elicit.org/datasets. Table 1 presents an overview of the datasets. More details are available in the Appendix.

**ADE Corpus V2 (*ADE*).** The ADE corpus V2 [15] contains sentences from medical case reports annotated for relation to adverse drug effects. We focus on the binary classification task of whether a sentence is related to an adverse drug effect (ADE).

**Banking77 (*B77*).** Banking77 [7] contains online banking customer service queries annotated with their intents.

**NeurIPS impact statement risks (*NIS*).** We include the broader impact statements from NeurIPS 2020 papers collected in the dataset from Ashurst et al. [1]. We annotate these based on whether they mention possibly harmful applications of the research done in the paper .[7]

---

[5] https://huggingface.co/datasets
[6] https://paperswithcode.com
[7] The raw scraped NeurIPS impact statements can be found at https://raft.elicit.org/neurips-impact.

| Dataset Name | Long inputs | Domain expertise | Detailed instructions | Number of classes | Test set size |
|---|---|---|---|---|---|
| ADE Corpus V2 (*ADE*) | – | ✓ | – | 2 | 5,000 |
| Banking77 (*B77*) | – | – | – | 77 | 5,000 |
| NeurIPS impact statement risks (*NIS*) | ✓ | – | – | 2 | 150 |
| OneStopEnglish (*OSE*) | ✓ | – | – | 3 | 516 |
| Overruling (*Over*) | – | ✓ | – | 2 | 2,350 |
| Semiconductor org types (*SOT*) | – | – | – | 3 | 449 |
| Systematic review inclusion (*SRI*) | ✓ | – | ✓ | 2 | 2,243 |
| TAI safety research (*TAI*) | ✓ | ✓ | ✓ | 2 | 1,639 |
| Terms of Service (*ToS*) | – | ✓ | ✓ | 2 | 5,000 |
| TweetEval Hate (*TEH*) | – | – | ✓ | 2 | 2,966 |
| Twitter complaints (*TC*) | – | – | – | 2 | 3,399 |

Table 1: Overview of the tasks in RAFT. *Long inputs*, *Domain expertise*, and *Detailed instructions* are some of the real-world challenges posed by RAFT.

**OneStopEnglish (*OSE*).** OneStopEnglish [31] contains articles sourced from The Guardian newspaper and rewritten by teachers to suit three levels of adult English as a Second Language (ESL) learners.

**Overruling (*Over*).** Overruling [39] contains statements from a law corpus annotated based on whether they are overruling, defined as nullifying a previous case decision as a precedent.

**Semiconductor org types (*SOT*).** We collect a dataset of institutions that have contributed to semiconductor conferences in the last 25 years, then classify these institutions into organization types: "university", "company", and "research institute".

**Systematic review inclusion (*SRI*).** We use data from a systematic meta-review studying interventions to increase charitable donations [22]. The task is to predict whether a paper advances past the screening stage.

**TAI safety research (*TAI*).** We include data from the formation of a bibliographic database for research on the safety of transformative artificial intelligence (TAI) [27]. We choose the binary task of predicting whether a work is classified as TAI safety research.

**Terms of Service (*ToS*).** The Terms of Service dataset [19] contains clauses from Terms of Services, annotated by whether they are potentially unfair to consumers.

**TweetEval Hate (*TEH*).** We include the hate-speech detection task from the TweetEval dataset [2], which was curated from Basile et al. [3].

**Twitter complaints (*TC*).** We include a dataset of tweets annotated by whether they contain a complaint [25].

### 3.2 Evaluation

#### 3.2.1 Setting and rules

The RAFT evaluation replicates real-world few-shot classification problems by restricting to 50 labeled examples without validation set, providing meaningful instructions and labels, and using a no-holds-barred setting:

**50 labeled examples.** We provide 50 labeled examples per task (not per class). In the authors' experience with users of the classification tool Elicit [23], this is approximately the number of examples people are willing to label for a task with a few thousand unlabeled examples. The 50

examples are chosen randomly, mirroring the applied setting in which one can't easily choose a balanced set. No examples beyond the chosen 50 are available for validation.[8]

**Task-specific instructions.** As an important replacement for large amounts of labeled data, instructions can specify how a task should be done. Therefore, we provide the instructions we give to human labelers so that they can be used in instructing automatic systems. The level of detail of the instructions varies. We write the instructions based on information from publications (for datasets published elsewhere) or in consultation with the dataset creator (for new datasets).

**Meaningful label names.** Similar to instructions, textual labels are an important aspect of few-shot and especially zero-shot learning. We create default textual labels for each dataset as recommended by FLEX [5].

**Transfer learning permitted.** Transfer and meta-learning using other datasets is permitted, including further pre-training on other corpora.

**Unlabeled data permitted.** Use of the unlabeled RAFT test sets is permitted, as unlabeled data are usually available in the applied setting.

**Open-domain retrieval permitted.** Models may be augmented with information retrieved from the internet, e.g. via automated web searches.[9]

**Submission requires only labels.** Submission on the test set is open to the public and only requires upload of test set labels. This is in line with benchmarks like GLUE [33] and SuperGLUE [32], but is in contrast to the few-shot benchmark FLEX [5]. By only requiring labels, we give submission creators maximal flexibility in what models to set up.

**Weekly evaluation.** Evaluation is run on a weekly basis to minimize information gained from frequent repeated submissions.

| Baseline | Avg | ADE | B77 | NIS | OSE | Over | SOT | SRI | TAI | ToS | TEH | TC |
|---|---|---|---|---|---|---|---|---|---|---|---|---|
| Human (crowdsourced) | **.735** | **.830** | **.607** | **.857** | **.646** | .917 | **.908** | .468 | .609 | **.627** | **.722** | **.897** |
| GPT-3 (175B) | .627 | .686 | .299 | .679 | .431 | **.937** | .769 | **.516** | **.656** | .574 | .526 | .821 |
| AdaBoost | .514 | .543 | .023 | .626 | .475 | .838 | .455 | .506 | .556 | .560 | .443 | .625 |
| GPT-Neo (2.7B) | .481 | .452 | .149 | .408 | .343 | .681 | .406 | .493 | .605 | .565 | .554 | .636 |
| GPT-2 (1.6B) | .458 | .600 | .121 | .561 | .245 | .498 | .380 | .492 | .612 | .498 | .311 | .723 |
| BART MNLI Zero-shot | .382 | .234 | .332 | .615 | .360 | .462 | .644 | .026 | .469 | .122 | .543 | .400 |
| Plurality class | .331 | .446 | .000 | .353 | .164 | .337 | .271 | .493 | .344 | .471 | .366 | .391 |
| GPT-3 Zero-shot | .292 | .163 | .000 | .572 | .323 | .378 | .628 | .027 | .362 | .164 | .303 | .290 |

Table 2: Performance of RAFT baselines (F1)

## 3.3 Metrics

Since some RAFT datasets have substantial class imbalances, we use F1 as our evaluation metric. We compute macro-averaged F1 scores, even for binary datasets. To get an overall score, we average across all datasets.

---

[8]By only releasing 50 labelled examples, we make it difficult to cheat by using more than 50 examples for validation. For the NeurIPS impact statement risks and Semiconductor org type datasets, the test set labels aren't available publicly. For other datasets, the test set labels are available publicly but it is non-trivial and discouraged to seek them out.

[9]Ideally, we'd only allow information from before the time each dataset needed to be labeled. This isn't currently feasible, so we settle for a fully open-domain setting.

# 4 Baselines

The code for all automatic baselines is open-sourced at https://raft.elicit.org/baselines.

## 4.1 GPT-3 baseline

We provide a simple automatic baseline using GPT-3 [6], accessed through the OpenAI API[10]. As in Brown et al. [6], we use in-context learning, adding labeled examples to the prompt to prime GPT-3. We also run a zero-shot version with no training examples included in the prompt.

### 4.1.1 Prompt construction

We build a prompt consisting of:

1. Task-specific instructions
2. $N$ labeled examples, with $N$ selected on a per-task basis
3. The target example to classify

Example prompts for all datasets are available at https://raft.elicit.org/baseline-prompts.

**Truncation.** GPT-3's context window support up to $2,048$ tokens. The experiments in Brown et al. [6] include as many complete labeled examples as would fit in the context, reporting that typically 10 to 100 examples fit. However, datasets such as *OSE* have very long inputs so that only 1-2 complete labeled examples would fit, suggesting that another approach may be better.

We select $N$ training examples to include in a given prompt and then truncate the examples. In a given task, the instructions take up $I$ tokens. Separators between the instructions and each example take up $S$ tokens. No more than $T = 2,048$ tokens can be used.

1. $E = T - I - S$ tokens are allotted for the training examples and classification target.
2. The classification target is truncated to $\frac{1}{4}E$ tokens.
3. Each of the $N$ remaining training examples is truncated to $\frac{3}{4}\frac{E}{N}$ tokens.

We truncate from a training example's data fields first, leaving the label intact.

**Field selection and sorting.** We exclude data fields that are unlikely to contribute substantially to GPT-3's performance. These fields either deal with the authors of the textual example or are URLs. Additionally, we sort the order in which the text fields occur to put the most important fields first. When examples are truncated, the most important information is preserved.

**Semantic selection.** To select training examples to include in the prompt for a given test example, we selected the most similar training examples as in Liu et al. [20]. To perform semantic search, we use the OpenAI API search endpoint with the `ada` engine.

### 4.1.2 Classification

With the prompt formed, we retrieve GPT-3's 100 most likely next tokens using the `davinci` engine. For each class, we assign the probability that its first token is generated. We then normalize the probabilities to sum to 1. For the *B77* dataset, multiple labels share the same first token so we prepend a numerical prefix such as "1. " to each class.

### 4.1.3 Parameter selection

We tune the GPT-3 baseline on the training set using *leave-one-out cross validation* (LOOCV): $k$-fold cross validation with $k = n$ so that only one test example is used at a time for validation. While LOOCV isn't robust with as few as 50 examples as discussed in Perez et al. [24], it is one of the best options for parameter selection in the few-shot setting. Detailed LOOCV results are in Section A.5.

**Instructions.** We test two modes of instruction: (a) a generic classification prompt: "Possible labels:" followed by a list of textual labels. (b) instructions similar to the ones given to human labelers, plus

---

[10]https://beta.openai.com

the list of textual labels. The instructions are taken whole when possible, and otherwise shortened and summarized manually to limit usage of the GPT-3 context window. Task-specific instructions outperform generic instructions by an $.04$ on averaged F1 score, thus we include task-specific instructions in the baseline.

**Semantic training example selection.** To select training examples for inclusion in the prompt from a larger set, we consider (a) selecting examples randomly and (b) using semantic search to identify the training examples most similar to the test example. Semantic selection outperforms random selection by $0.03$ on averaged F1, thus we include semantic selection in the baseline.

**Number of examples in the prompt.** We select the number of examples to include in the prompt on a per-dataset basis, as our truncation strategy induces a quality-quantity trade-off. For each dataset, we test performance with 5, 10, 25, and 50[11] training examples and choose the number that performs best by F1. For datasets with long inputs, smaller numbers of more detailed samples often produce better performance, while datasets with smaller inputs can fit more complete labeled examples in the prompt.

## 4.2 Other automatic baselines

**In-context baselines.** We run further in-context baselines GPT-Neo [4] and GPT-2 [26]. We provide code[12] for generating predictions on RAFT using these models and any other causal language model available on the HuggingFace Hub. For semantic search, we use a MiniLM [34] fine-tuned on sentence pairs via the sentence-transformers package[13].

**Zero-shot baselines.** We run two transformers in the zero-shot setting:

- GPT-3, to judge to what extent training examples in the prompt aid performance
- BART [18] trained on MNLI [36], as suggested by Yin et al. [38] and Davison [9] as an effective zero-shot classification approach

**Non-neural baselines.** We run AdaBoost [12] to establish a strong non-neural baseline. We construct feature vectors for each example based on the counts of $n$-grams of 1-5 words as the input to a weighted ensemble of 100 depth-3 decision trees. These decision trees and weights are trained with AdaBoost with learning rate 1, and evaluated through weighted voting. We also include a plurality (most frequent) class baseline.

## 4.3 Human baseline

To collect human baselines, we use the Surge[14] crowdsourcing platform. Following Wang et al. [32], we randomly select 100 data points from each test set and use a 2-step labeling process: qualification then annotation. The crowdsourced label is the plurality vote of 5 labelers.

We put crowd workers in a similar situation to automated systems. We link to a sheet with the same 50 labeled examples, use the same textual labels, and give the same task-specific instructions that we are providing to practitioners to adapt for instructing language models.[15]

## 4.4 Analysis

**Humans generally outperform GPT-3.** Humans outperform GPT-3 on 8 out of 11 tasks, demonstrating room for improvement for models on real-world few-shot tasks. We expect that exceeding the crowdsourced baseline will require substantial advances in model performance, and even more so for a future expert human baseline.

Weaknesses of GPT-3 include:

- **Many classes**: Humans most outperform GPT-3 on *B77*, which has by far the most classes in RAFT. With 77 classes and 50 labeled examples, many classes have no corresponding

---

[11]49 rather than 50 training examples for LOO experiments

[12]https://raft.elicit.org/baselines

[13]https://huggingface.co/sentence-transformers/all-MiniLM-L6-v2

[14]https://www.surgehq.ai/

[15]For details on the human baseline gathering process, see Section A.7.

labeled examples. Additionally, just listing out the possible classes takes up a large portion of GPT-3's context window.

- **Long inputs**: GPT-3 performs poorly on some tasks requiring reasoning over long inputs, such as *NIS* and *OSE*. GPT-3's context window may be a contributing factor.

**Crowd-sourced baselines struggle on domain-specific tasks.** Crowd-sourced humans substantially outperform GPT-3 on only 1 of 4 tasks we identified as requiring domain expertise:

- Humans substantially outperform GPT-3 on *ADE*, which requires medical expertise.
- Humans outperformed GPT-3 by just .053 on *ToS*, which requires parsing legal language.
- GPT-3 outperforms humans on *Over*, which requires greater legal expertise than *ToS* [39], and *TAI*, which requires expertise in AI safety research.

**Zero-shot performance is weak.** GPT-3 zero-shot does poorly on RAFT, performing worse than the plurality class baseline. BART zero-shot exceeds the plurality class baseline but does not do so in every dataset, and it is not competitive with few-shot language models. We encourage future research on improving performance in the zero-shot setting, perhaps through improved prompt construction and transfer learning.

**Neural baselines besides few-shot GPT-3 perform worse than AdaBoost.** Generative language models smaller than GPT-3 comfortably outperform the plurality class baseline but remain below AdaBoost. We use the same amount of labelled examples in the prompt as with GPT-3 despite the context window being smaller; performance may improve with fewer (but longer) examples.

## 5 Discussion

### 5.1 Limitations

**Linguistic diversity.** The benchmark only includes English tasks. Dealing with multilingual corpora is a real-world challenge for many NLP systems, especially for those deployed in countries where there are multiple national languages. To fully capture the distribution of real-world tasks, additional languages will be needed.

**Possible biases in data collection.** While we attempted to execute our dataset selection process as described in Section 3.1.3 in an unbiased manner, the datasets we ended up selecting are part of a subjective human process that may be subject to biases. For example, the organizations we work with are disproportionately in technology and policy.

### 5.2 Impact

**Offensive content.** By including a hate-speech detection dataset, we include offensive content and may harm readers of the dataset. We believe the advantages from studying hate-speech detection are likely greater than the disadvantages of publicizing hate-speech datasets.

**Prohibitive costs.** The models best equipped to perform well on RAFT will often be the massive transformer models trained by private corporations. In advancing this benchmark as a means of evaluating models, we risk further widening the gap between what a dedicated individual or team can do, and what can only be done by industry research labs with sufficient funding.

### 5.3 Future Work

**Stronger human baselines.** Human baselines are intended to tell us how well the dataset would be labeled in the absence of automated systems. For many RAFT datasets, this process would involve a stronger baseline than is easily available via a crowd-worker platform: for example, the *Over* dataset would be labeled by someone with law expertise. In addition to ML submissions, we welcome efforts to collect stronger human baselines for RAFT.

**Additional automatic baselines.** We expect that systems that use prompt-based fine-tuning rather than in-context learning may provide an even stronger automatic baseline. We further expect that models that leverage the open-domain information retrieval option can surpass models that don't.

**Application-specific metrics.** Different applications care about different metrics; e.g., in some applications it is more important to minimize false positives, whereas in others the focus is on false negatives. An ideal measure of real-world value would take that into account.

**Learning from natural language** In this work, we focused on instructions as a supplement to labeled examples. Similarly to Mishra et al. [21], we found that including task-specific instructions improved performance. Like humans, NLP systems could also learn from other types of natural language. For example, could including explanations with each labeled example be used to further improve few-shot performance?

## 6    Conclusion

RAFT is a benchmark that tests language models across multiple domains on economically valuable classification tasks in the true few-shot setting. To our knowledge, this is the first multi-task benchmark designed to closely mirror how models are applied in both the task distribution and the evaluation setup. By complementing existing synthetic benchmarks designed to highlight where models fall short, it helps measure the gap between research and practice, incentivizes work that is valuable for deployed systems, and provides a template for future benchmarks that mirror deployment.

## Acknowledgments and Disclosure of Funding

Our automatic baseline collection was subsidized by compute credits generously provided by OpenAI. Ethan Perez, Samuel Bowman, and Long Ouyang gave feedback on early versions of the RAFT concept and dataset lists. Douwe Kiela and Stella Biderman offered helpful advice on the project direction. Ross Gruetzemacher suggested inclusion of the Twitter Complaints dataset. We thank Thomas Wolf and Simon Brandeis for discussions and advice around the design of the benchmark's infrastructure.

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
