# OpenReview forum: "RAFT: A Real-World Few-Shot Text Classification Benchmark"
_NeurIPS.cc/2021/Track/Datasets_and_Benchmarks/Round2 — NeurIPS 2021 Datasets and Benchmarks Track (Round 2)_

### Official Review · Reviewer_J1AK · 2021-09-19
**Important problem**

**Rating:** 5
**Confidence:** 4
**Clarity:** The paper is well-written and easy to…

**Strengths:**

1. The paper is easy to follow, and the problem of benchmarking few-shot classification systems is well-motivated. The benchmark differs from existing few-shot text classification benchmarks.
2. The selection process and evaluation setting reflect real-world use cases.
3. The evaluation process is mostly well-conceived and polished up, e.g., the authors presumably took into account that without a development set, submitters will have an incentive to optimize hyperparameters on the test set (and thus introduced a weekly evaluation).

**Weaknesses:**

1. I do not expect papers that introduce benchmarks to place much effort on proposing the baselines. Nevertheless, the choice of GPT-3 as the only baseline in the classification task with 50 examples does not seem optimal to me. In particular, it is hard to assess how challenging are the proposed tasks given only GPT results – its performance might be anywhere between poor and excellent. The fact score is close to the human baseline does not help here because you relied on people without domain expertise (and thus, it is hard to tell what performance we should expect from experts in real-world application). I encourage authors to include an estimation of the score with the majority answer and some straightforward non-neural baseline. For example, XGBOOST can perform well with this number of examples on both short and long documents you tackle.
2.  I wonder how much the selection of these particular 50 documents impacts the overall score on the task, i.e., what would be the variance with the repeated random sub-sampling validation procedure. If low, then your more straightforward evaluation procedure with fixed training examples is justified. Nevertheless, such analysis was not conducted, and thus, I am concerned that there will be a significant noise level while comparing different solutions. It is hard to assess the benchmark utility without a similar analysis conducted with GPT-3 or another model.

**Additional Feedback:**

I do not consider it a weakness, but the number of examples seems arbitrary at the moment – even though you refer to "authors’ experience with users of the classification tool." If you are interested in a few-shot evaluation and 50 is a limit, why not provide other train sets with, e.g., 5, 10, and 25 examples? I expect people to use the benchmark in this manner anyway. Moreover, the inclusion of Banking77 with a larger number of classes than training examples is controversial – it is largely a zero-shot scenario where different architectures are supposed to work well.

Typo:
- There is a space before the dot in #144.

**Correctness:**

Newly-created datasets seem to be constructed in a sound way. Experiment design is probably correct.

**Documentation:**

Newly-created datasets are described in the Appendix. Baseline's source code is available on GitHub.

**Ethics:**

Several existing datasets included in the benchmark have unknown licenses and it is not clear whether they can be used.

**Relation To Prior Work:**

Differences are discussed in the introduction and Section 2.

**Summary And Contributions:**

The paper proposes a novel text classification benchmark consisting of real-world datasets with 50 training examples each. The main contributions are:
- review of available tasks and the selection process itself,
- annotation of a few novel datasets, provision of meaningful labels where required for existing ones,
- provision of human baseline estimates and GPT-3 baseline,
- preparation of evaluation platform.

---

> ### Author Response · Authors · 2021-09-28
> **Author Response**
>
> Thank you for your thoughtful review and suggestions!
>
> (1) More baselines
>
> We added the following baselines to the paper and leaderboard:
> * GPT-2
> * GPT-Neo
> * AdaBoost
> * BART MNLI zero-shot
> * Plurality label
> * GPT-3 zero-shot
>
> We find that GPT-3 significantly outperforms AdaBoost, a non-neural baseline. The difference between GPT-3 and AdaBoost is approximately the same as the difference between crowdsourced humans and GPT-3. However, smaller transformers perform poorer than AdaBoost. We hope that the provision of additional baselines makes the challenges posed by the RAFT benchmark clearer.
>
> (2) Lack of domain expertise
>
> In tasks such as Twitter Complaints, it would take relatively little time for a human to become a domain expert, while in Systematic Review Inclusion, it would be very challenging. Additionally, the tasks we selected are fairly diverse. Therefore, we could not easily train or find domain experts in all domains. We weren’t able to standardize domain-expert baselines, while crowdsourced baselines are identical across settings. We’re supportive of future work collecting stronger human baselines using domain expertise.
>
> (3) Noise from selection of 50 specific test examples
>
> Thank you for bringing up this concern. We most care about whether different sets of 50 training examples would change the relative ranking of various models, rather than whether there is variance within a single model. From the model rankings we have seen so far, we expect that the effect isn’t large (as the rankings approximately match what we would have expected).
>
> Additionally, this matches our target real-world setting. Without labels, there is little way to know which set of 50 examples should be labeled for optimal results. While structure in the dataset could be used, perhaps unsupervised learning to ensure diverse samples, in practice researchers are likely to select examples approximately at random.
>
> We would be very interested in seeing an analysis of this across multiple models. Unfortunately we aren’t able to include this analysis in this work due to the computational expense of many runs for large models.
>
> (4) Why not provide other training sets, e.g. 5, 10 and 25 examples?
>
> The main reason we focus on the 50-example setting is because we care about encouraging true few-shot learning, providing a clear separation between a public training set and private (or semi-private) test set. If we provided smaller training sets we would not be able to guarantee that people wouldn’t also use the larger ones.
>
> However, we’d support future work extending RAFT to include even lower resource settings. We would be happy to see submissions trained on subsets even for the current benchmark, and could provide filtered views on the leaderboard (but wouldn’t be able to enforce that people don’t use more examples than they claim).
>
> (5) Typo
>
> Thanks for pointing this out, it is fixed in the revised version.

---

### Official Review · Reviewer_wX7f · 2021-09-20
**A new few shot text classification benchmark**

**Rating:** 6
**Confidence:** 4
**Correctness:** Seems good.
**Clarity:** The paper overall is very well written.

**Strengths:**

- A few shot text classification dataset mimicking real world task.
- A carefully designed dataset with baseline result and leaderboard

**Weaknesses:**

- The main motivation for the dataset is not very clear as unlabelled test set in real world setting is easy to get.
- The zero shot baseline comparison is missing.
- For a new "domain" users will have to get 50 instances labelled. So, the dataset doesn't server the purpose.
- Few shot in low resource setting could have made much sense.
- Relatively small dataset

**Additional Feedback:**

what does authors mean by "representative of real-world classification tasks." is not very clear in section 2.2.

**Documentation:**

Yes, all details on data collection provided.

**Ethics:**

No.

**Relation To Prior Work:**

Yes, discussed.

**Summary And Contributions:**

Authors propose a new few shot text classification dataset and benchmark, RAFT. Authors claim that the proposed dataset is more realistic and mimics exact real world text classification task. Authors evaluation on RAFT reveals that existing text classification techniques struggle with RAFT dataset. Authors have also put together a leader board to track progress on RAFT.

---

> ### Author Response · Authors · 2021-09-28
> **Author Response**
>
> Thank you for your thoughtful review and suggestions!
>
> (1) Use of unlabelled test set
>
> We provide unlabelled test sets for each task in RAFT. We have clarified this in the paper.
>
> This is one of the ways in which our benchmark is distinct from some existing works (such as FLEX), which disallow use of the unlabelled test set.
>
> (2) Zero-shot baseline
>
> We’ve added both a GPT-3 generative zero-shot baseline and a BART MNLI zero-shot baseline. The GPT-3 baseline performs worse than the plurality class baseline and the BART MNLI baseline performs better than plurality class but worse than all few-shot baselines. We’d be interested in future work on improving them.
>
> (3) Requirement to get 50 labelled instances
>
> We have found that in applied settings with a few thousand examples to be classified, users are usually willing to label about 50 examples in a new domain. We initially considered a smaller # of labelled examples but found that less realistic because human workers are usually willing to trade 30 minutes of labelling for improved performance.
>
> 50 examples is close to (but arguably slightly more realistic than) the 32 labelled examples often used in prior work (e.g. [1], [2]).
>
> [1] https://arxiv.org/abs/2005.14165
>
> [2] https://arxiv.org/abs/2009.07118
>
> (4) Low-resource setting
>
> Submitters to RAFT are welcome to make submission that only rely on smaller #s of training examples, or submissions that rely on fewer resources. We’d be happy to provide filtered views on the leaderboard (with the caveat that we can’t enforce these constraints).
>
> (5) Dataset size
>
> The dataset is a comparable size to previously some of the current canonical few-shot benchmarks:
>
> * RAFT: 11 tasks, 28,712 total test examples
> * FewGLUE [1]: 8 tasks, ~17k total test examples
> * FLEX [2]: 20 tasks, 12 used for testing, ~175k total test set examples of which 114k are from one task
>
> [1] https://arxiv.org/abs/2009.07118
>
> [2] https://arxiv.org/abs/2107.07170
>
> (6) Representative of real-world tasks
>
> This refers to the distribution of text classification tasks currently carried out by humans to provide value in business and science settings, e.g. by office workers and research assistants.

---

### Official Review · Reviewer_kFZg · 2021-09-21
**Few-shot classification from real-world instructions, clear and straightforward setting**

**Rating:** 6
**Confidence:** 3
**Clarity:** The paper is well-written and easy to…

**Strengths:**

The most novel part of this dataset is including natural language instructions for the classification tasks. The authors presented several nice examples of such instructions, which are natural and reasonable in real-world scenarios.

As for the dataset itself, the data collecting procedure and the task settings are clear and sound.

**Weaknesses:**

The idea of including natural language task specification is interesting, but it seems that it is treated merely as a context in this paper. It is important to investigate these natural language instructions when constructing the dataset, since it is the most novel part of this dataset. For example, how different phrasing of the same classification task impact the performances are not investigated. Also, the instructions for different tasks in this dataset are completely separate. Given the same set of documents (e.g. semiconductor papers), we may ask the system to classify the institutions, or classify the papers by topics, etc. (otherwise it is not so meaningful to have natural language instructions). Another simple but interesting experiment to add is to test systems' performances on mismatched instruction/partially-labeled dataset, to investigate the true impact of task instructions (since all labels have natural language names, the performances might not be too bad).

From Table 2, it seems that the GPT-3 baseline already achieves reasonably good (or near-human) performances on 6 out of 11 tasks. Are these tasks too simple to be added to this dataset?



**Additional Feedback:**

Please see the weaknesses section.

**Correctness:**

The dataset is constructed in a sound way, and the baseline experiments and evaluation are designed appropriately.

**Documentation:**

The dataset is well-documented, and the authors provided the link to the dataset and its leaderboard.

**Ethics:**

No.

**Relation To Prior Work:**

Previous related works are discussed.

**Summary And Contributions:**

This paper proposes a real-world few-shot text classification benchmark, where the classification task instructions are in natural language (for the system to process) and the tasks are of real-world value (e.g. some task are difficult even for non-expert humans). The authors constructed their RAFT dataset by selecting 11 datasets across different areas. The authors designed clear and straightforward experimental and evaluation settings for this dataset. In the experiments, a GPT-3 baseline is implemented and tested against human performances.

---

> ### Author Response · Authors · 2021-09-28
> **Author Response**
>
> Thank you for your thoughtful review and suggestions!
>
> (1) Investigating the effect of instructions
>
> We strongly agree that this is a promising direction, and indeed one of the motivations for this benchmark. We encourage participants to experiment with different types of instructions. However, our goal is not to provide an optimal set of instructions. As the task instructions are drawn from a massive space of possibilities, we focused on setting up a rigorous evaluation environment that makes it easy to test different variations rather than do a necessarily very incomplete job at evaluating variations ourselves.
>
> (2) GPT-3 near human-level performance for some tasks
>
> First, we selected tasks to be representative of human work, whether they are difficult for ML models or not. We would like our benchmark to track to what extent few-shot NLP models can take the place of crowdsourced human work across a representative sample of tasks. This function requires that we include tasks even if current models already perform well on them.
>
> Second, we note in the paper that stronger human baselines could likely be achieved by using domain experts rather than crowd-sourced workers. The RAFT benchmark supports future work both on better ML models and on improved human expert submissions.

---

### Official Review · Reviewer_9Htn · 2021-09-21

**Rating:** 5
**Confidence:** 3
**Correctness:** This benchmark has appropriate evalua…
**Clarity:** This paper is well written.

**Strengths:**

- strong baseline (GPT-3)
- human performance is included to set a clear goal.
- integration with popular opensource NLP toolkit (huggingface)



**Weaknesses:**

- lack of baseline models
- most tasks are existing datasets

In my opinion, GPT-3 is not widely available to the research community. Researchers in universities often do not have the infrastructure to run experiments at GPT-3 scale (175B parameters). I would suggest adding more baseline scores from other pre-trained transformers such as BERT, GPT-2, Roberta, etc, so that the benchmark is more accessible to the research community.

Given that the only experimental result is from a private API (GPT-3 from OpenAI) and most of the tasks are existing benchmark, I think this paper does not make enough contribution.

**Additional Feedback:**

Include more accessible baselines such as GPT-2, RoBERTa would help reach a larger community.

**Documentation:**

Sufficient detail is included for reproducibility.

**Ethics:**

I don't find any ethical concerns in this submission.

**Relation To Prior Work:**

Related works are properly discussed.

**Summary And Contributions:**

This paper propose a benchmark of few-shot text classification. Recently, large scale pre-trained language models such as GPT-3 as demonstrated strong few-shot capability. The authors, however, argued that AI models' few-shot capability should be evaluated in "real-world" settings, and build an evaluation benchmark of 11 sub tasks. The study show that SOTA few-shot model (GPT-3) still underperforms non-expert human on 8 of the sub tasks.

---

> ### Author Response · Authors · 2021-09-28
> **Author Response**
>
> Thank you for your thoughtful review and suggestions!
>
> (1)  Accessible baselines
>
> We have added the following open-access baselines to the paper and leaderboard:
> * GPT-2
> * GPT-Neo
> * AdaBoost
> * Plurality label
> * BART MNLI zero-shot
>
> We also created this [starter kit notebook](https://github.com/oughtinc/raft-baselines/blob/master/src/raft_baselines/scripts/starter_kit.ipynb) which makes it trivial to generate and submit predictions for any causal language model on the [HuggingFace hub](https://huggingface.co/models?pipeline_tag=text-generation&sort=downloads) (e.g. XLNet, DistilGPT2, Reformer, GPT-J-6B).
>
> (2) Tasks from existing benchmarks
>
> Most previous few-shot benchmarks only gather existing datasets (e.g. FLEX [1], FewGLUE [2]). We believe that these benchmarks have been valuable for the community because they provide a target for ML work by assembling datasets with a common theme and standardizing data format and evaluation.
>
> Additionally, we go beyond most previous benchmarks by providing 2 new datasets (SOT and NIS), and 2 additional datasets not previously used in the NLP literature (TAI and SRI).
>
> [1] https://arxiv.org/abs/2107.07170
>
> [2] https://arxiv.org/abs/2009.07118

---

### Decision · Program_Chairs · 2021-10-11

**Decision:**

Accept

**Comment:**

This paper presents the RAFT which is a real-world few-shot text classification benchmark containing 11 sub tasks from different domains. This paper received borderline scores of (5,5,6,6). While all reviewers basically acknowledged that the dataset and the tasks are well-designed, the main concern was the lack of baselines other than GPT-3 (the result on GPT-3 itself is appreciated, though). Given the review, the authors updated the paper with the results on many other baseline models, which was appreciated and two reviewers agreed to raise the score from 5 to 6, although one of them is not reflected in the system. Thus all reviewers are now in the positive side.
Overall, there is not yet a particularly strong support for this paper, the AC considers it has no critical weakness and above the acceptance threshold.